

# The sex lives of ctenophores: the influence of light, body size, and self-fertilization on the reproductive output of the sea walnut, *Mnemiopsis leidyi*

Daniel A. Sasson[1,2] and Joseph F. Ryan[1,2]

[1] Whitney Laboratory for Marine Bioscience, University of Florida, St. Augustine, Florida, United States of America
[2] Department of Biology, University of Florida, Gainesville, Florida, United States of America

## ABSTRACT

Ctenophores (comb jellies) are emerging as important animals for investigating fundamental questions across numerous branches of biology (e.g., evodevo, neuroscience and biogeography). A few ctenophore species including, most notably, *Mnemiopsis leidyi*, are considered as invasive species, adding to the significance of studying ctenophore ecology. Despite the growing interest in ctenophore biology, relatively little is known about their reproduction. Like most ctenophores, *M. leidyi* is a simultaneous hermaphrodite capable of self-fertilization. In this study, we assess the influence of light on spawning, the effect of body size on spawning likelihood and reproductive output, and the cost of self-fertilization on egg viability in *M. leidyi*. Our results suggest that *M. leidyi* spawning is more strongly influenced by circadian rhythms than specific light cues and that body size significantly impacts spawning and reproductive output. *Mnemiopsis leidyi* adults that spawned alone produced a lower percentage of viable embryos versus those that spawned in pairs, suggesting that self-fertilization may be costly in this species. These results provide insight into the reproductive ecology of *M. leidyi* and provide a fundamental resource for researchers working with them in the laboratory.

## INTRODUCTION

Ctenophores (comb jellies) are fascinating planktonic animals most easily recognized by eight rows of fused cilia that they use as their primary means of locomotion. Recent work suggests ctenophores are the sister group to the rest of all animals and therefore are especially informative as to the state of the most recent common ancestor of animals (*Dunn et al., 2008*; *Hejnol et al., 2009*; *Ryan et al., 2013*; *Borowiec et al., 2015*; *Chang et al., 2015*; *Whelan et al., 2015*; but see *Pisani et al., 2015*). This phylogenetic position, the availability of nuclear and mitochondrial genome sequences (*Pett et al., 2011*; *Ryan et al., 2013*), and the ease with which embryos can be collected and observed (*Pang & Martindale, 2008b*) has made the ctenophore *Mnemiopsis leidyi* an emerging model

Corresponding author
Joseph F. Ryan,
joseph.ryan@whitney.ufl.edu

system for studying animal evolution and development (*Pang & Martindale, 2008a*). Furthermore, since the introduction of *M. leidyi* into European waters from its native Atlantic range (*Vinogradov et al., 1989*; *Reusch et al., 2010*) has had profound impacts on European fisheries (*Kideys, 2002*; *Oguz, Fach & Salihoglu, 2008*; *Finenko et al., 2013*), the biogeography and invasion ecology of *M. leidyi* continue to be important areas of study.

The reproductive biology and life-history of *M. leidyi* has likely played a major role in its ability to invade and establish populations in foreign waters. *Mnemiopsis leidyi*, like most ctenophores, are simultaneous hermaphrodites that have the ability to self-fertilize and have been observed to produce thousands of eggs a day (*Baker & Reeve, 1974*; *Costello et al., 2006*; *Graham et al., 2009*; *Jaspers, Møller & Kiørboe, 2011*; *Lehtiniemi et al., 2012*; *Jaspers, Costello & Colin, 2014*). Offspring may develop from egg to reproductive adult in as few as 13 days (*Baker & Reeve, 1974*; *Costello et al., 2012*). *Mnemiopsis leidyi* may even produce viable gametes as juveniles (*Martindale, 1987*).

A number of studies have described the spawning behavior of *M. leidyi* (*Baker & Reeve, 1974*; *Pang & Martindale, 2008b*). Earlier research suggested that *M. leidyi* spawns as a response to darkness (e.g., sunset, see *Freeman & Reynolds, 1973*), and a more recent study investigating the effects of starvation on egg production noted that during the summer, most eggs were produced over a 12 h dark period overnight from 19:00–7:00 (*Jaspers, Møller & Kiørboe, 2015*). However, the current spawning protocol for *M. leidyi* specifies that light cues trigger spawning, as gametes are readily released upon exposure to light after spending at least three to four hours in darkness (*Pang & Martindale, 2008b*).

Adult *M. leidyi* vary dramatically in body size, and this variation can affect both the likelihood to spawn and the number of eggs produced (*Baker & Reeve, 1974*; *Finenko et al., 2006*; *Graham et al., 2009*; *Jaspers, Møller & Kiørboe, 2011*). Animals are more likely to spawn as they grow larger (*Baker & Reeve, 1974*), and larger animals generally produce more eggs per day (*Baker & Reeve, 1974*; *Finenko et al., 2006*; *Jaspers, Møller & Kiørboe, 2011*; *Jaspers, Møller & Kiørboe, 2015*). However, the reported threshold size at which *M. leidyi* is able to spawn varies between studies, with some authors reporting smaller sizes of 10 and 15 mm (*Finenko et al., 2006*; *Jaspers, Møller & Kiørboe, 2011*), and some reporting thresholds as large as 32 mm (*Baker & Reeve, 1974*). In general, *M. leidyi* in European populations tend to spawn at smaller sizes (*Finenko et al., 2006*; *Jaspers, Møller & Kiørboe, 2011*) when compared to those in their native range (*Baker & Reeve, 1974*; *Graham et al., 2009*). It is unclear what factors are responsible for this wide variation in spawning-size threshold.

While it is true that self-fertilization provides the benefit of allowing *M. leidyi* to reproduce when conspecifics are not present, it may come with the cost of inbreeding depression. Inbreeding depression has been shown to affect the viability of offspring in many systems (*Charlesworth & Charlesworth, 1987*; *Crnokrak & Roff, 1999*; *Herlihy & Eckert, 2002*) such as snails (*Wethington & Dillon, 1997*) and adders (*Madsen, Stille & Shine, 1996*). Rates of self-fertilization and inbreeding depression may be especially high in recently established populations where the population size and genetic diversity are low (*Young, Boyle & Brown, 1996*; *Hedrick & Kalinowski, 2000*). Thus, establishing the degree to which self-fertilization is costly in *M. leidyi* has particular significance for the management

of areas where these ctenophores are invasive. However, to our knowledge, the costs associated with self-fertilization in *M. leidyi* have never been thoroughly investigated.

In this study, we aim to describe the spawning behavior, effect of body size on spawning, and potential costs of self-fertilization in *M. leidyi*. We first investigate spawning cues by placing individuals under different light regiments. We then describe how body size influences spawning likelihood, egg production, and egg viability. Finally, we test whether self-fertilization in *M. leidyi* is costly by comparing the viability of eggs from ctenophores spawned individually to those spawned with a partner. If self-fertilization is costly, we predict that the offspring of *M. leidyi* spawning alone will have lower viability than those spawned in groups. Taken together, this study provides a detailed description of the reproductive ecology of *M. leidyi,* adds new information for the management of nonnative ctenophores, and provides an important resource for establishing *M. leidyi* as a model system in the laboratory.

## MATERIALS AND METHODS

### Collection

We collected a total of 218 *M. leidyi* for the following experiments between June and October 2015 from the surface waters of Port Orange and St. Augustine, Florida using a cteno-dipper (beaker on a stick) between the hours of 9:00 and 15:00. We generally collected animals on sunny days with low winds. We then transported them in buckets to the Whitney Laboratory for the Marine Biosciences in St. Augustine, FL. Upon arrival, the ctenophores were transferred first to a large beaker with filtered seawater. All seawater used in the experiments was pumped to the laboratory directly from the ocean and filtered with a 0.2 μm filter. The temperature of the seawater ranged from 25–29.5 °C although water temperatures likely acclimated to room temperature during experiments (see below). The salinity of the seawater ranged between 35 and 36 ppt. We measured the polar length of every ctenophore along the oral/aboral axis to the nearest mm using calipers and placed them in individually marked 4″ diameter glass dishes filled with 250 mL of filtered seawater. Ctenophores were used in the experiments the same day of collection except for 19 individuals that were kept overnight and used the following day (see below). Following the experiments, we released all ctenophores except for four individuals which were used for DNA and RNA extraction for another study. Due to their short time period in the lab, we did not feed any ctenophores. All experiments were conducted at room temperature (range 20–25 °C).

### Ctenophore distribution across experiments

The ctenophores we collected were often used in multiple analyses when appropriate. The 38 ctenophores used to test the *Pang & Martindale (2008b)* protocol were not used in any other analysis. All the remaining *M. leidyi* that we collected were placed individually in bowls (N = 118) and were used to measure the effect of body size on spawning likelihood. Since we did not refine our egg estimation protocol (see below) until partway through the experiment, we measured the correlation between body size and egg production using 30 *M. leidyi*. Of these 30, we measured egg viability 24 h later in all

but one individual. Finally, we compared egg production and offspring viability between the aforementioned 30 ctenophores and an additional 50 *M. leidyi* that were placed in bowls in pairs (N = 25 bowls). For more information on which ctenophores were used in each experiment, see the Supplemental Data.

## Light effects on spawning

We tested the protocol described in *Pang & Martindale (2008b)* using 38 *M. leidyi* that we collected between June 30, 2015–August 12, 2015 (see Supplemental Data). These ctenophores ranged in size from 27–57 mm (median: 43 mm). These ctenophores were either placed in the experiment the day of collection (N = 19) or kept overnight in a large kreisel aquarium and placed in the experiment the day following collection (N = 19). Between the hours of 10:00 and 18:00, we placed these animals in 4″ dishes with 250 mL of filtered seawater in the dark for three to four hours. Upon exposure to light, bowls were monitored over the next two hours for the presence of eggs.

We conducted a separate set of experiments to test the importance of light cues for spawning on a subset of the *M. leidyi* that we collected from Port Orange and St. Augustine between August 20, 2015 and September 14, 2015 (N = 66, size range: 19–58 mm, median: 41 mm). On the day of collection, we separated each ctenophore into individual 4-inch diameter bowls filled with 250 mL of filtered seawater and haphazardly assigned individuals to one of four treatments: A) constant light (N = 21), B) 11 h of light and then four hours of darkness (N = 15), C) seven hours of light and then eight hours of darkness (N = 12), or D) constant darkness (N = 18). For the variable light treatments, 7 or 14-watt compact fluorescent bulbs were attached to an automatic timer that turned off after the set amount of time. The animals in the constant light treatment were placed under a lamp with a 15-watt compact fluorescent bulb. All treatments began at 18:00 and ended at 9:00 the next day, at which point we exposed all of the animals to light and immediately recorded whether eggs were present in each bowl. In this experiment, we did not count the number of eggs spawned in each bowl, as we had not yet developed our egg counting protocol (see below).

## Size effects on spawning, egg production, and egg viability

In many systems, body size strongly influences reproductive output. We designed an experiment to test the effect of body size on spawning likelihood, egg production, and offspring viability. We tested the effect of size on spawning likelihood using the ctenophores already spawned in the previous light cues experiment (N = 66) and an additional 52 *M. leidyi* (total N = 118) that we collected from Port Orange and St. Augustine between September 16, 2015 and October 15, 2015. We measured the length of every ctenophore along the oral/aboral axis to the nearest mm using calipers and then placed each in their own bowl with 250 mL of filtered seawater. To ensure spawning, we left the additional 52 animals overnight in either constant darkness for 15 h (N = 26) or in a room with no artificial lights and an uncovered window to experience natural changes in light (N = 26). We immediately recorded whether eggs were present in each bowl on the following morning at 09:00. Since *M. leidyi* typically spawn hundreds of eggs,

we only considered bowls with at least 25 eggs as representing a true spawning event. We calculated the effect of size on spawning likelihood using logistic regression and visualized the data with a cubic spline.

A number of the ctenophores produced thousands of eggs, making a direct count of all eggs difficult. To address this challenge, we developed a protocol to allow us to estimate the number of eggs in each two-inch bowl. We drew a two-inch diameter circle and placed a square within the circle so that each point on the square touched the edge of the circle (Fig. 1). Finally, we divided the square into eight equal sized triangles that we labeled 1–8. For each ctenophore, we counted the number of eggs in two randomly selected triangles. Two triangles comprise 15.91% of the total area of the circle, and so to estimate the total number of eggs in the dish we multiplied the combined egg count by 6.285. We tested this protocol by comparing the estimated number of eggs to the actual number of eggs produced by two ctenophores with lower egg counts. The number of eggs estimated was close enough to the actual number of eggs (50 estimated vs. 42 actual and 31 estimated vs. 29 actual) that we felt that this measure provided us with at least a way to compare relative egg production across individuals. To collect the eggs of the ctenophores that spawned, we poured the water and eggs from each bowl through a 70-$\mu$m filter. The eggs of each ctenophore were then pipetted into separate two-inch diameter bowls filled with filtered seawater. Eggs were allowed to settle in the bowl before we counted eggs. Estimated egg production was log-transformed to increase normality. We then evaluated the correlation between body size and estimated egg production using linear regression for the individuals that spawned (N = 30). Egg production from more *M. leidyi* was not included in this analysis because we developed this method of estimating egg production halfway through the study.

To determine egg viability, we re-counted the number of eggs in each dish after 24 h. *M. leidyi* typically develop into juvenile cydippids within 24 h after fertilization between 18 and 20 °C (*Martindale & Henry, 1997*). Juveniles can easily be distinguished from undeveloped eggs due to ciliary movement, and since viable embryos can swim away from their original triangle into the water column, we counted the number of undeveloped eggs in the same triangles as in the egg production assay. We then estimated the number of undeveloped eggs in the entire dish using the method described above. Using this estimate we calculated the percent of undeveloped eggs (estimated undeveloped eggs/estimated total eggs) and subtracted that number from one to determine the percentage of viable eggs. We used linear regression to assess the effect of body size on egg viability (N = 29).

## Costs of self-fertilization

If self-fertilization is costly, we would expect *M. leidyi* spawning alone to have reduced offspring viability compared to those spawning in pairs. To test for such a cost, 80 *M. leidyi* collected from Port Orange and St. Augustine between September 7, 2015 and October 15, 2015 were randomly placed alone or with another individual in a 4″ diameter bowl with 250 mL of filtered seawater. Individuals spawned overnight and the next morning we estimated the number of eggs present in each bowl and the percent of viable

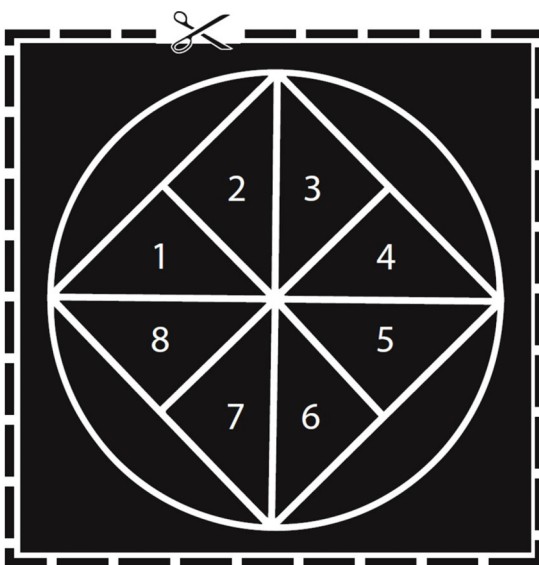

**Figure 1 Diagram used to estimate egg numbers in *M. leidyi*.** Each triangle (labeled 1–8) represents 7.96% of the total area of the circle. We counted the eggs in two triangles and then multiplied the total by 6.285 to estimate the total number of eggs in the dish. Scaled to actual size used for round glass bowls 2″ in diameter.

offspring 24 h later (see above). We compared estimated egg production and egg viability from ctenophores spawned alone (N = 30 for egg production, N = 29 for egg viability as we accidently did not count one bowl for viability) to ctenophores spawned in pairs (N = 25) using Student's t-test.

All statistical analyses were run in JMP 11.0 (SAS Institute, Cary, NC).

## RESULTS

### Spawning light cues

Following the recent spawning protocol (*Pang & Martindale, 2008b*), only three of 38 (7.9%) animals produced any eggs. Furthermore, the few ctenophores that did spawn often released only a few eggs (median = 19 eggs, range 18–177 eggs).

When placed in bowls overnight, we found no difference in spawning likelihood between ctenophores kept in constant light (20/21 [95%] spawned), four hours of darkness (15/15 [100%] spawned), eight hours of darkness (12/12 [100%] spawned), or constant darkness (17/18 [94%] spawned).

### Size effects on spawning and egg viability

The ctenophores in this experiment varied in size from 12–70 mm (median = 38 mm). As *M. leidyi* grew larger, the likelihood of spawning significantly increased (Fig. 2, Logistic regression, N = 118, $\chi^2$ = 62.0, p < 0.0001). All but three ctenophores larger than 30 mm spawned overnight, while only one ctenophore smaller than 26 mm produced eggs.

We saw substantial variation in the number of estimated eggs spawned (range = 25–3934 eggs, median = 484 eggs). Larger individuals generally produced more eggs (Fig. 3, N = 30, $r^2$ = .38, p < 0.001). The light conditions the ctenophores experienced

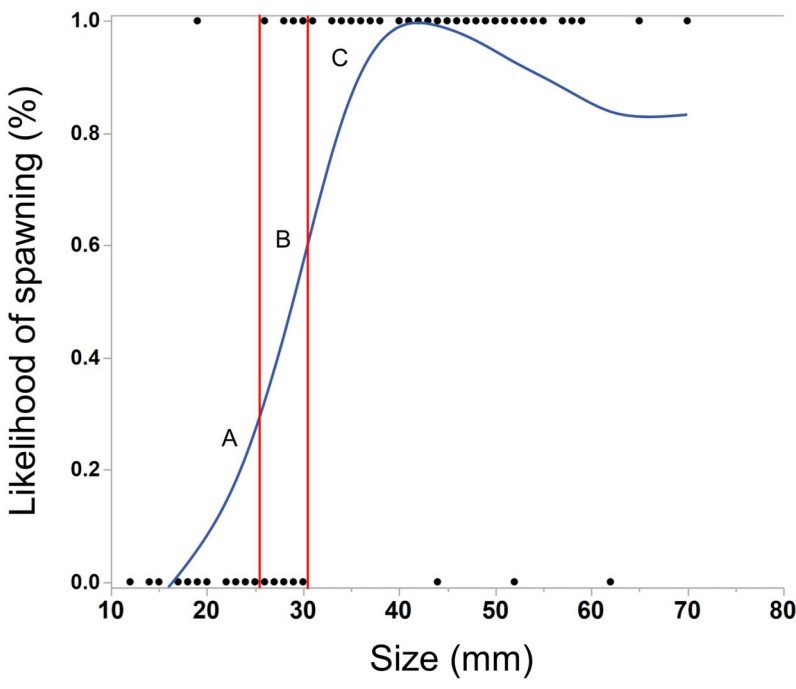

**Figure 2 Cubic spline showing the effect of body size of *M. leidyi* on the likelihood to spawn.** Points along the lower x-axis indicate individuals that did not spawn, while points on the upper x-axis indicate individuals that did spawn. Multiple individuals of the same size may be represented by a single point. Ctenophores smaller than 26 mm (section A) rarely spawned (1/22 = 5%) while those larger than 30 mm (section C) almost always spawned (77/80 = 96%). Nearly half of the individuals between 26 and 30 mm spawned (section B, 6/16 = 38%). Lambda value of cubic spline set to 1.

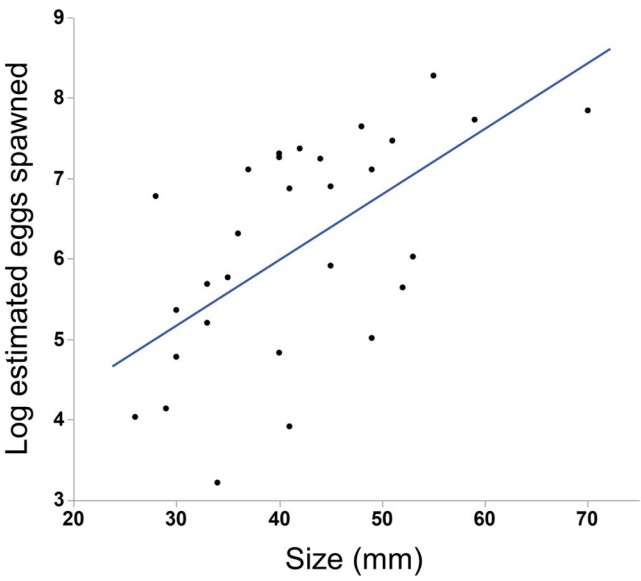

**Figure 3 Effect of body size on egg production in *M. leidyi*.** Larger individuals generally produced more eggs than smaller individuals (N = 30, $r^2$ = .38, p < 0.001). Only those animals that spawned 25 or more eggs are included in the analysis and figure.

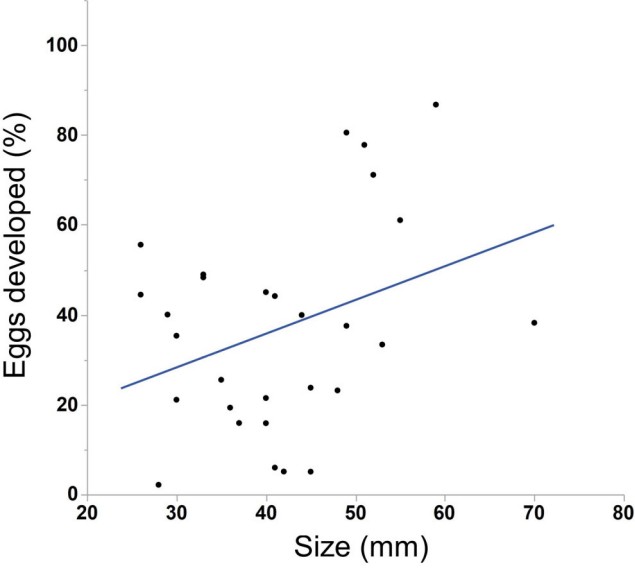

**Figure 4 Correlation between body size and egg viability in *M. leidyi*.** Body size positively correlated with the percentage of eggs that developed after 24 h, although the result was not significant (N = 29, $r^2$ = 0.12, p = 0.07).

overnight did not affect egg production (ANOVA, $F_{5,28}$ = 1.45, p = 0.24). We also found a weak but insignificant positive correlation between body size and egg viability (Fig. 4, N = 29, $r^2$ = 0.12, p = 0.07).

## Costs of self-fertilization

We compared the egg production between *M. leidyi* that spawned alone (N = 30) with *M. leidyi* that spawned in pairs (N = 25). We found no difference between treatments in the estimated number of eggs produced (Fig. 5, Student's t-test, t-ratio = 0.005, p = 1.0). However, we did find that a higher percentage of offspring from individuals that spawned in pairs (N = 25) had developed after 24 h when compared with individuals that spawned by themselves (N = 29, Fig. 6, Student's t-test, t-ratio = 2.3, df = 52, p = 0.025).

## DISCUSSION

Previous work has suggested that *M. leidyi* uses light cues to induce spawning (*Freeman & Reynolds, 1973*; *Pang & Martindale, 2008b*; *Martindale & Henry, 2015*). However, our attempts at replicating this spawning cue failed; few *M. leidyi* placed into the darkness during daytime hours spawned and those that did spawn produced few eggs. Instead, we found that almost every *M. leidyi* over 30 mm spawned overnight regardless of the light/dark cycle despite the slight differences in light intensity used in these experiments; even those individuals that were placed under constant light consistently spawned. This result suggests that these *M. leidyi* spawned using a circadian rhythm rather than specific light cues. We have identified the circadian rhythm genes *Clock* and ARNT in the *M. leidyi* ML2.2 gene models (*Moreland et al., 2014*) using a Maximum-likelihood approach (see supplementary figure). These and other circadian rhythm genes have been associated with reproduction and reproductive timing in a number of systems (*Boden &*

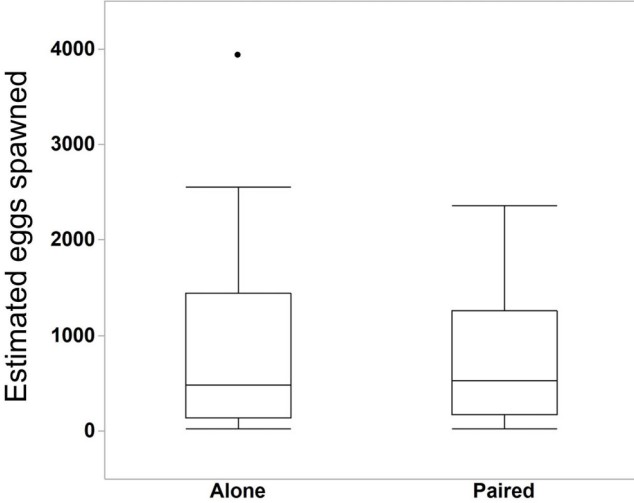

**Figure 5 Estimated number of eggs in bowls of individuals that spawned alone (N = 30 bowls) and in pairs (N = 25 bowls).** Surprisingly, two *M. leidyi* spawning together did not produce more eggs than individuals spawning alone (Student's t-test, t-ratio = 0.005, p = 1.0). The data point above the "Alone" box plot indicates an individual that spawned an estimated 3,934 eggs. Removing this data point did not change the overall findings of the analysis.

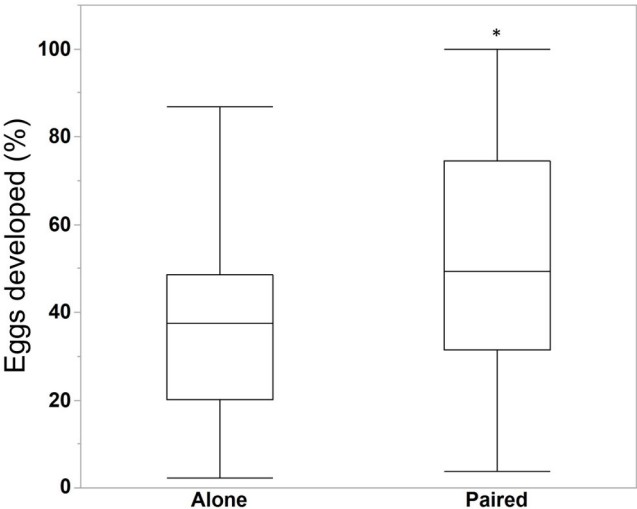

**Figure 6 Percentage of eggs developed after 24 h for individuals *M. leidyi* spawning alone (N = 29 bowls) and in pairs (N = 25 bowls).** A higher percentage of eggs developed for *M. leidyi* in pairs, possibly suggesting a cost to self-fertilization (Student's t-test, t-ratio = 2.3, df = 52, p = 0.025). Asterisk indicates significant difference across treatments.

*Kennaway, 2006*; *Leder, Danzmann & Ferguson, 2006*; *Liedvogel et al., 2009*). Functional genomic analyses into how these circadian-rhythm genes affect spawning could potentially provide solid evidence linking circadian rhythms and *M. leidyi* spawning. Given the phylogenetic position of ctenophores as the sister lineage to the rest of animals (*Dunn et al., 2008*; *Ryan et al., 2013*; *Borowiec et al., 2015*; *Chang et al., 2015*;

*Whelan et al., 2015*; but see *Pisani, et al., 2015*), such a study might also address to what extent the genetic circuitry underlying animal circadian rhythm was present in the last common animal ancestor.

Previous spawning protocols were described for *M. leidyi* populations near Woods Hole, Massachusetts (*Pang & Martindale, 2008b*). To our knowledge, spawning protocols have not previously been described for *M. leidyi* in the Atlantic waters of northern Florida. While these two *Mnemiopsis* populations had previously been classified as separate species (Massachusetts = *Mnemiopsis leidyi*, A. Agassiz 1865, northern Florida = *Mnemiopsis mccradyi*, Mayer, 1900), they are now generally considered to be separate populations of the same species (*Pang & Martindale, 2008a*; *Bayha et al., 2015*), although this has yet to be extensively tested genetically. Populations within species may differ in their reproductive timing or cues (e.g. *Partecke, Van't Hof & Gwinner, 2004*; *Moore, Bonier & Wingfield, 2005*; *Jaspers, Møller & Kiørboe, 2011*) and thus it could be that the spawning behavior we observed is unique to the northern Florida population of *M. leidyi*. Alternatively, spawning behavior could change across seasons with changes to daytime length or water temperature (e.g. *Sastry, 1963*; *Fell, 1976*; *Taranger et al., 1998*).

Body size plays an essential role in ctenophore reproduction. Spawning occurs almost exclusively in larger *M. leidyi* (>30 mm), although a few individuals smaller than 30 mm spawned and a few animals larger than 40 mm did not spawn (Fig. 3). Interestingly, this result differs from *M. leidyi* reproduction in the Caspian and Baltic Seas where individuals commonly spawn when over 10 mm and the most common size of spawning individuals is between 20 and 30 mm (*Finenko et al., 2006*; *Jaspers, Møller & Kiørboe, 2011*). Why these populations differ in size of reproduction is unclear, but they may be influenced by water temperature, resource abundance, or low salinity (*Finenko et al., 2006*; *Jaspers, Møller & Kiørboe, 2011*; *Jaspers, Møller & Kiørboe, 2015*). The differences in the non-native *M. leidyi* might also be a result of selection for body size or age of reproductive maturity due to selective pressures imposed by ship-ballast transport. The size of spawning may also change seasonally. This study took place from June to early October when water temperatures in Florida are high. Studies investigating the spawning behavior of Atlantic *M. leidyi* across seasons and water temperatures would be informative.

Not surprisingly, larger individuals in our study produced more eggs than smaller individuals (Fig. 4). Body size may correspond to nutritional status rather than age (*Reeve, Syms & Kremer, 1989*) and so larger ctenophores may simply be those well fed enough to produce gametes. The production of gametes is costly (*Hayward & Gillooly, 2011*) and smaller ctenophores preferentially allocate resources to somatic growth rather than gamete production (*Reeve, Syms & Kremer, 1989*). Since larger individuals consume more prey (*Bishop, 1967*; *Finenko et al., 2006*) they likely have more resources available to produce eggs than smaller individuals.

Body size may also affect offspring viability. We found that the percentage of developed eggs after 24 h increased as individuals grew larger (Fig. 4), although this result was marginally not significant. If body size truly does affect the number of viable offspring produced it may be due to the volume of sperm available to a particular individual.

If sperm are limited, especially in small individuals, larger animals may simply have more sperm available to fertilize eggs. Alternatively, larger animals may provision more resources to their eggs than smaller animals, which may increase egg viability or development speed. This possibility could be tested by comparing the size of eggs across body sizes.

We also found that *M. leidyi* individuals spawning alone had a lower percentage of developed offspring after 24 h than ctenophores that spawned in pairs (Fig. 6). What contributes to this apparent cost to self-fertilization is unclear. It could be that spawning pairs simply fertilize more eggs than individuals spawning alone, which might occur if sperm are limited. Another possibility could be that the percentage of eggs fertilized did not differ between treatments but that fewer fertilized eggs developed for individuals spawning alone. Although we did not differentiate between unfertilized eggs and non-developing embryos in this study, we did commonly observe embryos that appeared to have arrested development after only a few stages of cell division in both treatments. These results are consistent with a reduction in offspring viability due to inbreeding depression.

Interestingly, ctenophores in pairs did not produce more eggs than those spawning alone (Fig. 5). The average size of the ctenophores did not differ between treatments, suggesting that, when paired, *M. leidyi* either reduced the number of eggs spawned or only one of the two ctenophores spawned eggs. This latter option may indicate the intriguing possibility that *M. leidyi* alternates between releasing sperm and eggs when in pairs or groups. This phenomenon, known as egg-trading, has been reported in other simultaneously hermaphroditic systems including sea slugs, tobacco fish, and polychaetes (*Leonard & Lukowiak, 1984*; *Sella, 1985*; *Petersen, 1995*). This behavior could be used to reduce the chance of self-fertilization in *M. leidyi*. However, the underlying assumption of egg-trading is that individuals spawn with the same partners multiple times; we would not expect this to be the case in *M. leidyi* under natural circumstances since movement is largely governed by water flow.

While ctenophores spawning alone had decreased offspring viability, our results also suggest that these individuals may be more efficient than when spawning in pairs. Since paired *M. leidyi* did not spawn more eggs than individuals that spawned alone, more total viable offspring were produced per individual for those that spawned alone despite their reduced offspring viability. This result may actually suggest a benefit to spawning alone. However, these results should be cautiously interpreted as we only spawned each ctenophore once. Gamete production is costly (e.g. *Hayward & Gillooly, 2011*), and since *M. leidyi* that spawned alone released more gametes per individual than paired *M. leidyi*, they likely require a longer refractory period for gametogenesis before spawning again. Thus this initial increase in total viable offspring may only be temporary and continued self-fertilization may prove detrimental over multiple spawning events. Comparing the reproductive output and viability between paired and single individuals over multiple days would provide more resolution on the costs associated with self-fertilization.

The ability to self-fertilize almost certainly enhances the capability of ctenophores to spread when undergoing range expansion. However, the potential costs to self-

fertilization that we have demonstrated in this study may at least slow down or limit their ability to establish new, low-density populations. These costs may be especially high at the initial stages of an introduction when population numbers and genetic diversity are low. Our self-fertilization experiment only examined one stage of development (i.e., 24 h after spawning) in one generation and yet we still found evidence that self-fertilization is costly. Additional costs likely do not appear until later in life or after multiple generations of self-fertilized offspring. An experiment investigating the multi-generational effects of self-fertilization may provide a clearer picture of the reproductive constraints, or lack thereof, that *Mnemiopsis* populations experience when initially expanding into new geographic areas.

## CONCLUSIONS

Due to their evolutionary position as the sister lineage to all other animals (*Ryan et al., 2013*), ctenophores in general, and *M. leidyi* in particular, are quickly emerging as new model systems from which to understand evolution, development, regeneration, and even human disease (*Pang & Martindale, 2008a*; *Maxwell et al., 2014*). Understanding the reproductive ecology of ctenophores is a necessary step in establishing these animals as tractable models for these areas of research. This study has reinforced the importance of body size in *M. leidyi* reproduction and has provided the first suggestions that self-fertilization may be costly in ctenophores. However, ctenophore reproduction in natural systems is still very much a mystery. For example, little is known about how common it is for *M. leidyi* to self-fertilize in the wild. We have shown that spawning likely follows a circadian rhythm, which may be a mechanism to increase the odds of out-crossing if all animals spawn simultaneously. If self-fertilization is indeed costly, additional adaptions to increase the chance of out-crossing are likely. This work provides a fundamental resource for researchers working with *M. leidyi* in their laboratory, as well as a foundation from which future studies of *M. leidyi* reproductive biology can be launched.

## ACKNOWLEDGEMENTS

We acknowledge Leslie Babonis, Kira Carreira, Marta Chiodin, Bailey Steinworth, and Allison Zwarycz for help with collecting *Mnemiopsis leidyi*. We thank Mark Martindale and David Simmons for advice on ctenophore husbandry and spawning. We thank Scott Santagata for his comments on previous versions of this manuscript. Finally, we would like to thank two reviewers for their helpful comments and suggestions, which greatly improved this manuscript.

### Funding

This work was funded with startup funds to Joseph Ryan from the University of Florida DSP Research Strategic Initiatives and the Office of the Provost. This material is based

upon work supported by the National Science Foundation under Grant No. (1542597). The funders had no role in study design, data collection and analysis, decision to publish, or preparation of the manuscript.

## Grant Disclosures
The following grant information was disclosed by the authors:
National Science Foundation: 1542597.

## Competing Interests
The authors declare that they have no competing interests.

## Author Contributions
- Daniel A. Sasson conceived and designed the experiments, performed the experiments, analyzed the data, wrote the paper, prepared figures and/or tables, reviewed drafts of the paper.
- Joseph F. Ryan conceived and designed the experiments, performed the experiments, contributed reagents/materials/analysis tools, wrote the paper, reviewed drafts of the paper.

## Data Deposition
The raw data has been supplied as a Supplemental Dataset.

## Supplemental Information
Supplemental information for this article can be found online at http://dx.doi.org/10.7717/peerj.1846#supplemental-information.

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
