# Peer review of "The sex lives of ctenophores: the influence of light, body size, and self-fertilization on the reproductive output of the sea walnut, Mnemiopsis leidyi"

_PeerJ, doi:10.7717/peerj.1846_

## Round 0.1 · original submission · Major Revisions

· Academic Editor

Major Revisions

I examined your manuscript and both reviewers' comments in making my decision of Major Revision.

In addition to the reviewer's comments, I have read through the manuscript myself, and have added a few comments of my own:

1. The References section needs to be formatted properly, please take care of this.
2. Some of the figures need work:
a. A scale for Figure 1 would be highly useful for readers, and the legend as of now provides no information.
b. Why is there a scissors symbol around the edge of Figure 2?

Although this is a Major Revision, I feel most comments should be able to be addressed easily. Please address both reviewers' comments, both sets of comments are pertinent to improving the paper.

·

Basic reporting

The text is generally clear and uses unambiguous text.
However, please rephrase the Materials and Methods section, as "to be spawned" tends to refer to the second generation (born of eggs produced during experimentation), which is not the case in this study. Use instead the term "to spawn" or "to produce eggs".
Also, I do not think the cubic spline illustrated in Figure 3 is necessary. There are clearly 3 sections to the graph (<26mm = 0% spawn; 26-30mm = x% spawn; >30mm = 100% spawn). You could simply add vertical lines to the graph and indicate the percentage of animals that spawned.

Experimental design

No Comments.

Validity of the findings

No Comments.

Additional comments

In the reference section, the species mane Mnemiopsis Leidei should be in italics.

Reviewer 2 ·

Basic reporting

The text is very well written and enjoyable to read. The article includes sufficient introduction and background to demonstrate how the work fits into the broader field of knowledge. While the authors introduce some of the previous work conducted with reproduction of Mnemiopsis leidyi, however, some of the relevant literature is missing. E.g. in work conducted by Jaspers et al. (several papers), who have previously demonstrated the effect of salinity, body size and prey concentration (starvation) on egg production of Mnemiopsis leidyi. Thus, statements such as "the threshold size before spawning begins has varied from 15mm (Finenko et al. 2006) to 32mm (Baker & Reeve 1974) across studies" should be rewritten. Some minor detailed comments related to this in attachment.

Experimental design

The submission clearly defines the research question, which is relevant and meaningful. The authors have identified the knowledge gap being investigated (self-fertilization costs), and demonstrate how their study contributes to filling this gap. However, in the current form, there are methodological procedures which are questionable and which can alter the results and conclusions. Thus, some more detailed information on the methods parts should be described with more sufficient information e.g. light conditions of the sampling, salinity, temperature, time between sampling and experiments, were the experimental specimens fed, what was the filter size...and if not included these should be at least taken up in the discussion as a potential error source (effect of starvation on the body size and egg production). Some minor detailed comments related to this are in attachment.

Validity of the findings

Despite the overall strengths, to my opinion, the quality of the data is hard to determine as there are several questions related to the methods, and thus results and findings were difficult to interpret. Most likely, the answers to the questions do not change the results, but will give more validity to the findings.

Additional comments

The topic is relevant and important given that this gelatinous organism has tendency for being invasive species, for high reproduction and potential being model organism. Thus information on factors affecting to their reproduction and development is needed. While I am happy to say that this is an interesting and well written paper, there are some problems that warrant attention prior to publication. I have attached some comments/questions/recommendations to the manuscript PDF.

Annotated reviews are not available for download in order to protect the identity of reviewers who chose to remain anonymous.

---

## Round 0.2 · Minor Revisions

· Academic Editor

Minor Revisions

Both reviewers felt this new version was much improved. However, reviewer 2 still has one concern that needs to be addressed, and both reviewers have some small corrections to be made. Thus, my decision is "minor revisions".

·

Basic reporting

The authors correctly addressed the points raised by the reviewers, and have improved the readability of the manuscript. I recommend this manuscript for publication with only a few very minor changes (see below).

Experimental design

No Comments.

Validity of the findings

No Comments.

Additional comments

Introduction l.45 and l.50: Mnemiopsis should be written in full when at the beginning of a sentence.

Materials and Methods l.148: change "been spawned" to "spawned".

Results l.208: variations

Reviewer 2 ·

Basic reporting

Where as I find that the manuscript has improved greatly, I still have some minor concerns which should be addressed prior to publication (which I recommend). My main concern is in the ambiguous description of the results and discussion on the cost of self-fertilization. In the current form, you state that there was 1) no difference on number of eggs spawned whether spawned alone or paired, 2) statistical difference on egg development on favor of being spawned in pairs (Fig 6), but then you also state that: " more total viable offspring were produced per individual when spawned alone than when paired despite their reduced percentage of offspring that developed" which could be interpret that there is no cost of self-fertilization. This part needs to be clarified.

Experimental design

No comments

Validity of the findings

No comments

Additional comments

Minor comments:
Line 55: Be precise with terms, here with night. Which time of the year? How many hours of darkness?
Lines 56-57: Are there more protocols than just Pang & Martindale, 2008b?
Line 72: Provide an example of such systems
Line 80 and throughout the text: You use the wording “reproductive cues”, while you only test the effect of light (not salinity, temperature etc.). Change the wording.
Lines 94-96: Enough with: “We generally collected animals on sunny days with low winds”
Line 100: Add also the water temperature, and salinity doesn’t have ppt.
Line 124: No need to mention feeding (you already state that above)
Lines 119- 127: I still abit confused which samples where used for what experiment. Maybe it could clarify is you would use experiment 1, 2 and so on. Here you provide dates when these specimens were collected, but in lines 128-140 you only provide location. Provide same information for all the experiments.
Lines 148-150: why different setting? To ensure the spawning?
Lines 196-198: Why such a low percentage of spawning? The size of these specimens were mostly above the threshold.
Lines 223-225: Not in the first experiment.
Line 224: Provide the size
Line 225: Maybe just state slight difference in light intensity
Line 228: Blast search in GeneBank?
Lines 270-271: Please provide information on how much the percentage increased (from what to what).
Lines 272-274: How does the sperm volume effect to the offspring viability? It might effect on number of fertilized eggs, but to offspring viability? Could you provide some references to base this speculation?
Fig 3: Why 25 or more eggs? In text you use 15
Fig 5 and 6: Are numbers in paired analyzed per individual or per bowls? Needs to be clarified.

---

## Round 0.3 · accepted · Accept

· Academic Editor

Accept

The revision was well done, and there are no major outstanding issues with this manuscript. I have gone over it in detail myself, and have added some small English corrections here and there in the attached PDF file. Please ensure these are done at the proof stage if not earlier.